# RANDOMIZED TRIALS IN EEG CLASSIFICATION EXPERIMENTS ARE MANDATORY AS DRIFT IS PERVASIVE

## ABSTRACT

Temporal correlation is demonstrated in three public EEG datasets. Filtering does not remove the correlation. The community is cautioned that a substantial number of recent publications are flawed due to a confound between temporal correlation and stimulus class, despite claims that filtering removes the correlation. The only known way to avoid this confound is through randomization.

## 1 INTRODUCTION

A recent sequence of papers (Spampinato et al., 2016; 2017; Kavasidis et al., 2017; Palazzo et al., 2017; 2018; Tirupattur et al., 2018; Palazzo et al., 2020a;b; 2021; 2024) claims to classify visual stimuli from EEG recordings. With the exception of Tirupattur et al. (2018), which uses a dataset from Kumar et al. (2018), these all use a dataset from Spampinato et al. (2017), an early version of which was reported in Spampinato et al. (2016). Li et al. (2021), Ahmed et al. (2021; 2022), and Bharadwaj et al. (2023), however, demonstrated a confound in these datasets: a spurious correlation between stimulus class and time, due to poor experiment design. This invalidates the work reported in nearly one hundred papers as discussed below. Recent responses (Palazzo et al., 2020b; 2024) argued that

   I) the design of the data-collection effort in Li et al. (2021) and Ahmed et al. (2021) induced or exacerbated the temporal correlation due to long recording sessions that may introduce fatigue, and

   II) the bandpass filtering performed by Spampinato et al. (2017) Kavasidis et al. (2017), and Palazzo et al. (2017; 2018; 2020a;b; 2021; 2024) could remove the temporal correlation.

Here, we demonstrate that

A) three public datasets with short recording sessions also contain a temporal correlation and
B) bandpass filtering does not remove the temporal correlation.

This supports the view that temporal correlation is pervasive in EEG data and that the only known valid way to avoid a confound is through appropriate randomization.

## 2 SIGNIFICANCE

Several independent lines of research have refuted a large body of flawed work (Spampinato et al., 2017; Kavasidis et al., 2017; Palazzo et al., 2017; 2018; 2020a;b; 2021; 2024) along completely different axes. Li et al. (2021) demonstrated that the dataset used (Spampinato et al., 2017), and the methods used to collect that dataset, suffer from a temporal confound, correlating stimulus class with experiment timing. Accuracy drops to chance when the confound is removed. Ahmed et al. (2021) demonstrated that this holds even with a much larger dataset. Ahmed et al. (2022) demonstrated that this holds for the additional classifiers used in Palazzo et al. (2018; 2020a;b; 2021). Bharadwaj et al. (2023) demonstrated that this holds even when using supertrials.

Here we progress beyond prior demonstrations that the particular dataset (Spampinato et al., 2017) is confounded. We offer the novel claim that the collection protocol is inherently and irreparably confounded; any dataset collected with this protocol would by its very nature suffer from the same confound. This responds specifically to claims I and II above with our counter claims A and B in ways that have not previously been addressed. This is significant because the confounded dataset

continues to be used, and confounded collection methods continue to be used to collect new confounded datasets, justified, in part, by claims I and II. This is further significant for the following reasons:

- Nearly one hundred papers (An & Cho, 2016; Spampinato et al., 2016; Ben Said et al., 2017; Bozal Chaves, 2017; Kavasidis et al., 2017; Palazzo et al., 2017; Parekh et al., 2017; Spampinato et al., 2017; Zhang et al., 2017; Du et al., 2018; Fares et al., 2018; Kumar et al., 2018; Palazzo et al., 2018; Piplani et al., 2018; Tirupattur et al., 2018; Wang et al., 2018; Zhang & Liu, 2018; Zhang et al., 2018; Zhong et al., 2018; Du et al., 2019; Hwang et al., 2019; Jiang et al., 2019; Jiao et al., 2019; Long et al., 2019; Mukherjee et al., 2019; Uys, 2019; Wang et al., 2019; Cudlenco et al., 2020; Fares et al., 2020; Li et al., 2020; Palazzo et al., 2020a;b; Wang et al., 2020; Zheng et al., 2020a;b; Palazzo et al., 2021; Zheng & Chen, 2021; Ma et al., 2021; Mo et al., 2021; Jiang et al., 2021; Lee et al., 2021; Cavazza et al., 2022; Khaleghi et al., 2022; Lee et al., 2022; Mishra et al., 2022; Mishra, 2022; Scharnagl & Groth, 2022; Shimizu & Srinivasan, 2022; Ahmadieh et al., 2023; Bai et al., 2023; Du et al., 2023; Duan et al., 2023; Hasan & A, 2023; Imani et al., 2023; Lan et al., 2023; Lee et al., 2023; Liu et al., 2023; Singh et al., 2023; Song et al., 2023; Wahengbam et al., 2023; Zeng et al., 2023b;a; Fan et al., 2024; Ferrante et al., 2024a;b; Gou et al., 2024; Lei et al., 2024; Liu et al., 2024a;b; Luvsansambuu et al., 2024; Mishra et al., 2024; Mwata-Velu et al., 2024; Ngo et al., 2024; Palazzo et al., 2024; Pan et al., 2024; Qian et al., 2024; Singh et al., 2024; Tang et al., 2024; de la Torre-Ortiz et al., 2024; Yang & Liu, 2024; Ye et al., 2024; Zheng et al., 2024b;a; Zhu et al., 2024; Deng et al., 2025; Fares, 2025; Fu et al., 2025; Lopez et al., 2025; Mehmood et al., 2025; Singh et al., 2025; Xiang et al., 2025) draw flawed conclusions based on the confounded dataset from Spampinato et al. (2017) and datasets suffering from the same confound.
- A number of new datasets have been collected with this same confounded protocol (Gou et al., 2024; Pan et al., 2024; Zhu et al., 2024; Qian et al., 2024; Uys, 2019; Shimizu & Srinivasan, 2022; Liu et al., 2024b; Wang et al., 2019; 2020; Ma et al., 2021; Cudlenco et al., 2020; Zheng et al., 2024b; Cavazza et al., 2022; Luvsansambuu et al., 2024; Liu et al., 2023; Bai et al., 2023; Parekh et al., 2017).
- A number of these have been publicly released and are used by others. For example, Singh et al. (2023), Singh et al. (2024), and Lopez et al. (2025) use the dataset reported in Kumar et al. (2018) and Duan et al. (2023), Singh et al. (2024), and Lopez et al. (2025) use the dataset reported in Ma et al. (2021).
- This is further egregious because Palazzo et al. (2020b; 2024) continue to claim that their dataset (Spampinato et al., 2017), and their results that were obtained with that dataset (Spampinato et al., 2017; Kavasidis et al., 2017; Palazzo et al., 2017; 2018; 2020a;b; 2021; 2024), are valid, despite the refutations in Li et al. (2021), Ahmed et al. (2021; 2022), and Bharadwaj et al. (2023), in part, because of claims I and II in Palazzo et al. (2020b; 2024).
- This has been used to justify continued publication of a large and growing body of flawed work based on confounded datasets (Cavazza et al., 2022; Khaleghi et al., 2022; Lee et al., 2022; Mishra et al., 2022; Mishra, 2022; Scharnagl & Groth, 2022; Shimizu & Srinivasan, 2022; Ahmadieh et al., 2023; Bai et al., 2023; Du et al., 2023; Duan et al., 2023; Hasan & A, 2023; Imani et al., 2023; Lan et al., 2023; Lee et al., 2023; Liu et al., 2023; Singh et al., 2023; Song et al., 2023; Wahengbam et al., 2023; Zeng et al., 2023b;a; Fan et al., 2024; Ferrante et al., 2024a;b; Gou et al., 2024; Lei et al., 2024; Liu et al., 2024a;b; Luvsansambuu et al., 2024; Mishra et al., 2024; Mwata-Velu et al., 2024; Ngo et al., 2024; Palazzo et al., 2024; Pan et al., 2024; Qian et al., 2024; Singh et al., 2024; Tang et al., 2024; de la Torre-Ortiz et al., 2024; Yang & Liu, 2024; Ye et al., 2024; Zheng et al., 2024b;a; Zhu et al., 2024; Deng et al., 2025; Fares, 2025; Fu et al., 2025; Lopez et al., 2025; Mehmood et al., 2025; Singh et al., 2025; Xiang et al., 2025) even after the confound became known through the work of Li et al. (2021), Ahmed et al. (2021; 2022), and Bharadwaj et al. (2023).

Current machine-learning conferences, and more generally, computer-science conferences and journals, are loathe to publish refutations. Observing this, Schaeffer et al. (2025) proposed that the field of machine-learning establish a "refutations and critiques" track in prominent conferences. While we applaud and support this proposal, the current lack of such a track should not be an impediment to publishing refutations. Scientific journals in other fields have long done so, often resulting in retraction of flawed work. Schaeffer et al. (2025) offer five example pieces of claimed flawed work in machine learning. Each is an individual paper. These pale in comparison to the flaws we uncover

here: a systemic flaw of the entire peer review process across an entire field of inquiry, namely classification of stimulus image class from EEG recordings, that affects seventeen datasets and ninety one papers. Moreover, none of the five examples in Schaeffer et al. (2025) are egregious; here the authors of the flawed work continue to argue for its validity despite four refereed refutations and fifty new flawed papers have been published subsequent to these four refereed refutations. This argues for the need to make the community aware of the severity of the issue.

## 3 METHOD

The dataset from Spampinato et al. (2017) recorded EEG data from six subjects, in four runs per subject, each run containing ten blocks, each block presenting fifty stimuli for 0.5 s each, with 10 s blanking between blocks. All fifty stimuli in each block were of the same class and the data from all subjects were pooled and divided into training and test sets, leading to the confound. We replicate this design with public EEG data as closely as possible. The twist is that two datasets (Hatlestad-Hall et al., 2021; Babayan et al., 2019) contain resting-state data and the third (Delorme, 2020) contains data where all trials have the same stimulus. We construct a simulated experiment design, repurposing and repartitioning this data in a fashion that mirrors the design of Spampinato et al. (2017) as closely as possible, associate block-level class labels with this data in the manner of Li et al. (2021, § 3.7), and train and test the LSTM classifier from Spampinato et al. (2017) and the EEGChannelNet classifier from Palazzo et al. (2018; 2020a;b; 2021; 2024) on this repurposed data. Above-chance classification accuracy, both with and without bandpass filtering, demonstrates the temporal correlation in this data and the fact that filtering fails to remove that correlation.

The design of Spampinato et al. (2017) is $6 \times 4 \times (10 \times (50 \times 0.5 \text{ s} + 10 \text{ s}) - 10 \text{ s})$, requiring 340 s of data per run and 1360 s per subject. Hatlestad-Hall et al. (2021) contains 4 min resting-state data for 111 subjects. Due to insufficient data per subject, we pooled multiple (four) subjects into a single virtual subject (six total), taking each subject's data as a run. The 4-min runs allowed simulating 50 stimulus presentations for each of 4 runs with 7 classes per run, for a total of 28 simulated classes, for each of 6 subjects: $6 \times 4 \times (7 \times (50 \times 0.5 \text{ s} + 10 \text{ s}) - 10 \text{ s})$. Babayan et al. (2019) contains 16 min of resting-state data for each of 216 subjects, organized as sixteen 60 s blocks, alternating between eyes closed and eyes open. We used only the first five eyes-closed blocks for each subject. We took each 60 s block to simulate two blocks from the design of Spampinato et al. (2017): $50 \times 0.5 \text{ s} + 10 \text{ s} + 50 \times 0.5 \text{ s}$, took five such blocks from each subject to simulate ten blocks from the design of Spampinato et al. (2017), pooled the data from four subjects to simulate each virtual subject, for a total of six virtual subjects: $6 \times 4 \times 5 \times (50 \times 0.5 \text{ s} + 10 \text{ s} + 50 \times 0.5 \text{ s})$. This allowed simulating 50 stimulus presentations for each of 20 runs with 2 classes per run, for a total of 40 simulated classes, for each of 6 subjects. Delorme (2020) contains three runs, each with 13 min of data, for each of 13 subjects, divided into 1 s trials. Each trial contained one of 3 stimuli: 70% contained the same tone A, 15% contained the same tone B, and 15% contained white noise. We extracted only the 70% of the trials with tone A, concatenated them into a simulated run of 546 s, divided this into 10 simulated blocks with 50 simulated stimulus presentations per block and 10 s blanking between blocks, pooled the data from two subjects to create each virtual subject, with each virtual subject having four virtual runs, two from each subject, for a total of six virtual subjects: $6 \times 4 \times (10 \times (50 \times 0.5 \text{ s} + 10 \text{ s}) - 10 \text{ s})$. This allowed simulating 50 stimulus presentations for each of 4 runs with 10 classes per run, for a total of 40 simulated classes, for each of 6 subjects.

Note that two of the datasets are resting state. The third is completely auditory. Thus none of the datasets required visual attention during the recording sessions, and two did not require any attention at all.

## 4 RESULTS

For each dataset, the simulated data for 6 subjects were pooled and randomly split into 80% training and 20% test data, with results averaged across five such splits. This was repeated twice, first without any filtering (Figs. 1 and 2 row 1 for LSTM, row 3 for EEGChannelNet) and then with 14–71 Hz bandpass filtering (row 2 for LSTM, row 4 for EEGChannelNet).[1] Note that without filtering,

---

[1]Results were obtained using the code for LSTM and EEGChannelNet from https://github.com/perceivelab/eeg_visual_classification, modified slightly to support z-scoring, plotting the

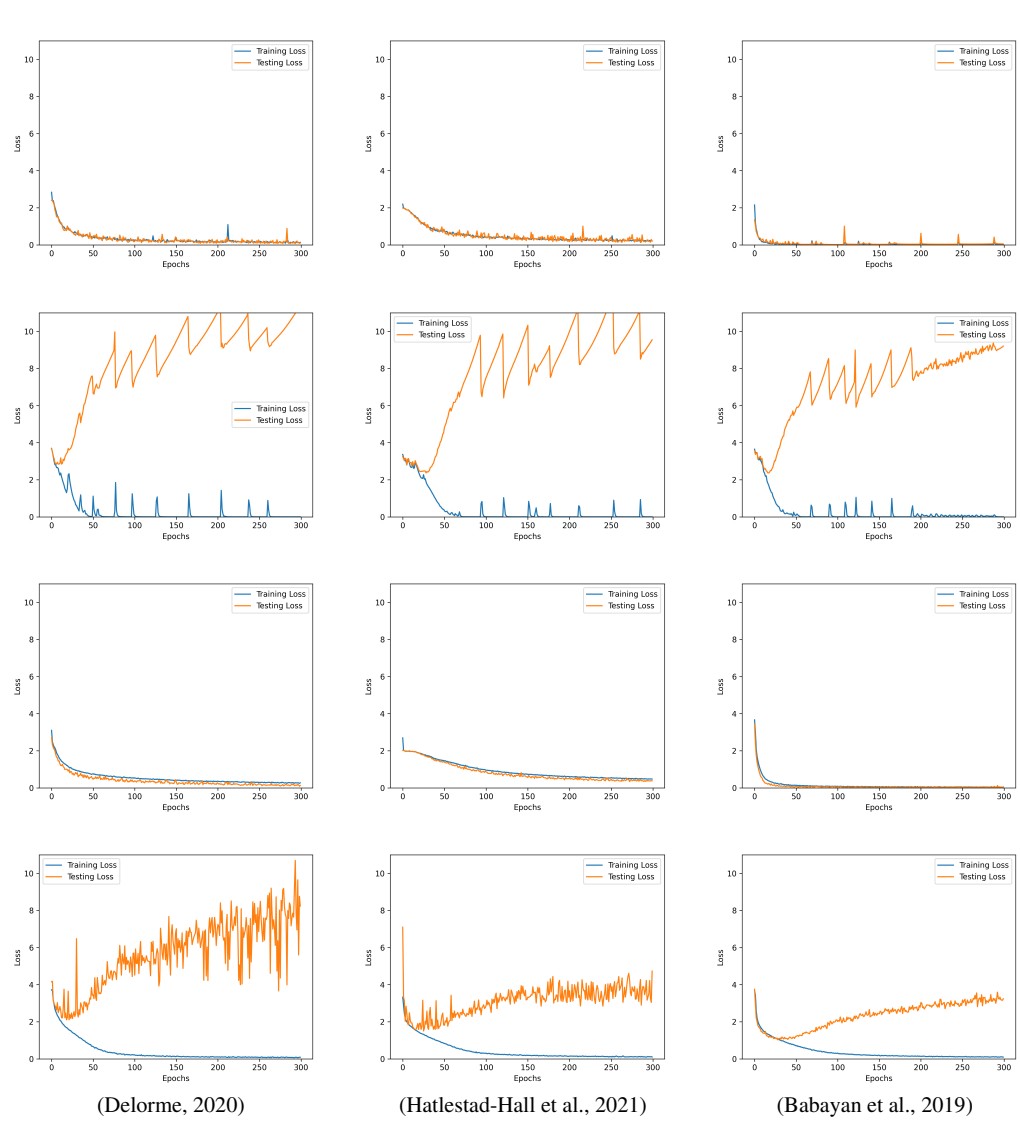

(Delorme, 2020)                (Hatlestad-Hall et al., 2021)                (Babayan et al., 2019)

Figure 1: Plots of loss *vs.* epoch for the three datasets, for the LSTM classifier (rows 1 and 2) and the EEGChannelNet classifier (rows 3 and 4), averaged over splits, without filtering (rows 1 and 3) and with filtering (rows 2 and 4). Training in blue, testing in orange.

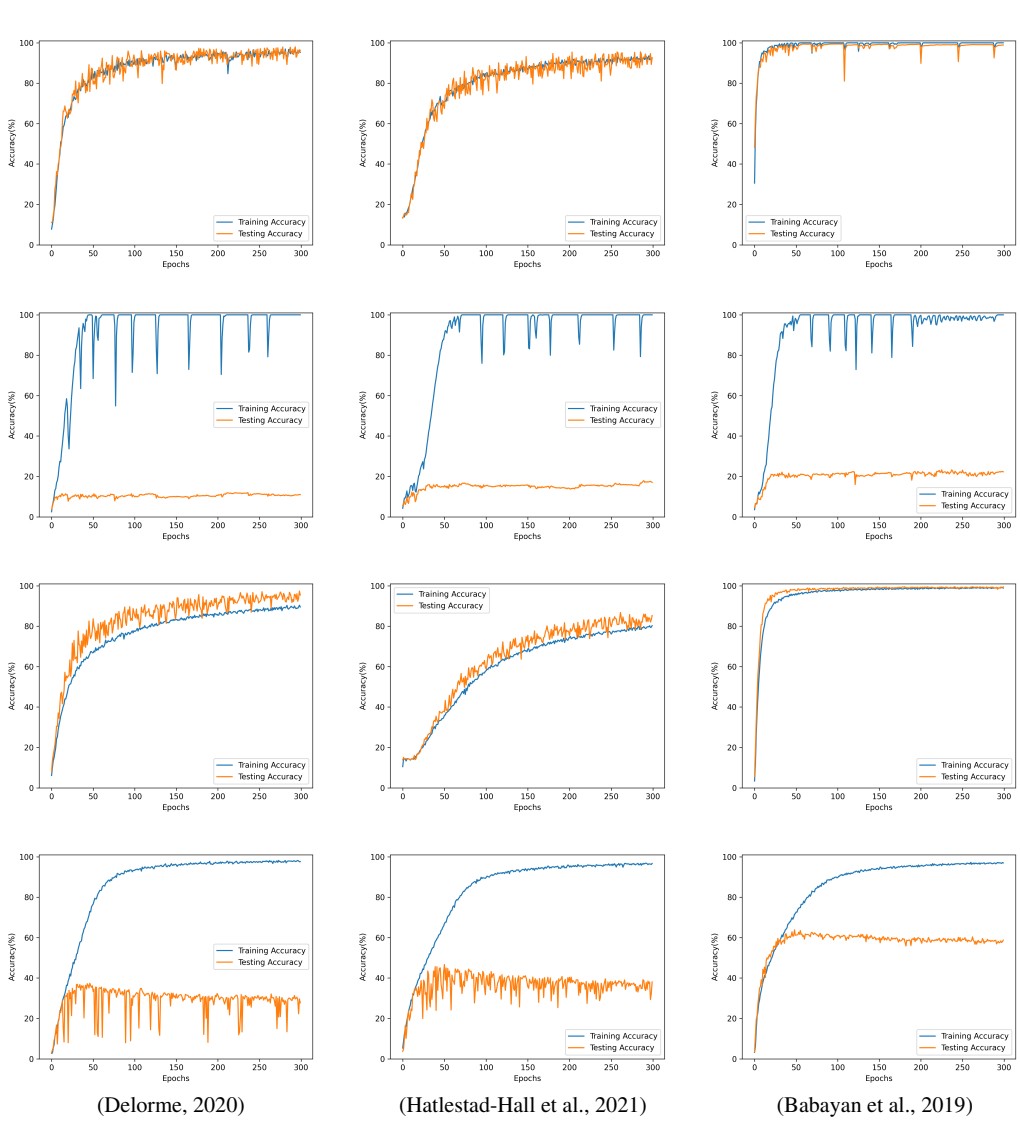

(Delorme, 2020)        (Hatlestad-Hall et al., 2021)        (Babayan et al., 2019)

Figure 2: Plots of accuracy *vs.* epoch for the three datasets, for the LSTM classifier (rows 1 and 2) and the EEGChannelNet classifier (rows 3 and 4), averaged over splits, without filtering (rows 1 and 3) and with filtering (rows 2 and 4). Training in blue, testing in orange.

classification accuracy is near perfect for both LSTM and EEGChannelNet. While classification accuracy is much lower with filtering, it is still far above chance (2.5% for Delorme 2020 and Babayan et al. 2019, 3.6% for Hatlestad-Hall et al. 2021). Also note that with filtering, EEGChannelNet yields considerably higher classification accuracy than LSTM. This suggests that EEGChannelNet particularly focuses on the confound, even more so than LSTM. This, together with prior results demonstrating chance performance of LSTM and EEGChannelNet on nonconfounded data (Li et al., 2021; Ahmed et al., 2021; 2022; Bharadwaj et al., 2023), suggests that LSTM and EEGChannelNet focus particularly on the confound, and not the stimulus-related brain activity, calling all results in Spampinato et al. (2017), Kavasidis et al. (2017), and Palazzo et al. (2017; 2018; 2020a;b; 2021; 2024) into question.

## 5 DISCUSSION

¡The results clearly demonstrate that these three public datasets exhibit a temporal correlation. Note that we make no claim that the studies underlying these datasets exhibit a confound; we are clearly using these datasets for a different purpose than they were intended. As pointed out by Li et al. (2021), one must be careful when repurposing data not to introduce a confound. We do so here only to demonstrate that temporal correlation appears to be pervasive in EEG data, not just the data collected in Spampinato et al. (2017). By design, our repurposing demonstrates that it exists even in short runs. The strong correlation between training and test, for both loss and accuracy, as a function of epoch throughout the entire training regimen clearly indicates label leakage.

The results also clearly demonstrate that the temporal correlation is not removed by bandpass filtering. High-pass filtering removes low-frequency components, not slow spectral change. One way to understand this is the fact that broadcast radio works by modulating an HF carrier with an audio-frequency signal, tuning a receiver with a bandpass filter to a specific channel, and demodulating the filtered broadcast to recover the audio-frequency signal. This, perhaps, is the most foundational principle of Electrical Engineering.

An extensive body of work is flawed because of this confound. Not only is all work using the dataset from Spampinato et al. (2017) flawed, new datasets (Gou et al., 2024; Pan et al., 2024; Zhu et al., 2024; Qian et al., 2024; Uys, 2019; Shimizu & Srinivasan, 2022; Liu et al., 2024b; Wang et al., 2019; 2020; Ma et al., 2021; Cudlenco et al., 2020; Zheng et al., 2024b; Cavazza et al., 2022; Luvsansambuu et al., 2024; Liu et al., 2023; Bai et al., 2023; Parekh et al., 2017) have been collected with the same confounded design. Ma et al. (2021) states:

> to improve the problem of low signal-to-noise ratio of EEG signals, the same
> category of images were presented successively to bring continuous stimulation to
> the brain

Bharadwaj et al. (2023) demonstrates how it is possible to increase the signal-to-noise ratio of EEG signals and achieve better classification accuracy without introducing a confound due to temporal correlation.

## 6 CONCLUSION

It appears that temporal correlation is pervasive in EEG data, even for short runs, where the subject is not fatigued. When conducting experiments with EEG data, one must be careful not to let temporal correlation introduce a confound. Temporal correlation cannot be removed *post hoc* by filtering. The only known way of preventing temporal correlation from introducing a confound is through appropriate randomization that breaks the temporal correlation.

AUTHOR CONTRIBUTIONS

Removed for blind review.

---

loss and accuracy as a function of epoch, and these datasets which have 62 and 64 channels instead of the original 128.

ACKNOWLEDGMENTS

Removed for blind review.

ETHICS STATEMENT

This work debunks nearly one hundred published papers whose results are based on the same confound: a correlation between stimulus class and temporal drift. This confound has been found in eighteen available EEG datasets. Just as with an inconsistent set of axioms one can prove anything, a confounded dataset can be used to support any claim, even ones that are false or absurd. That is what many recent publications based on this confound do: things like generating high fidelity renderings of images, or even 3D CAD models of objects, from EEG recordings.

A research community, knowingly or unknowingly, has discovered that one can use confounded datasets to churn out a plethora of flawed results without reviewers noticing. They have also discovered that one can collect new confounded datasets to churn out even more flawed results without reviewers noticing. The temptation to do this is so strong that the community continues to do so four years after details of the confound were published.

It is conceivable that the flaws in these datasets may be a driving factor behind their frequent reuse. When a dataset is severely confounded, it becomes relatively easy to achieve an extremely high accuracy, which can in turn be used to support sensational claims, and ultimately directs further attention to the dataset. In business, this phenomenon is referred to as "the bad money drives out the good money."

More prominent exposure of these flawed methods and consequent false results will allow resources wasted on continued use of these confounded datasets and flawed methods to be reallocated. The debunked work also causes direct ongoing harm:

- grant proposals can be rejected due to preliminary results not being competitive with results demonstrating falsely-inflated performance based on confounded data or faulty methods;
- manuscripts can be rejected for the same reason;
- grants can be awarded based on false pretenses
- manuscripts can be accepted for the same reason;
- degrees can be awarded for the same reason;
- resources can be wasted attempting to replicate the debunked results;
- resources can be wasted having people read and review flawed papers, and learn flawed methods; and
- because the debunked work relates to brain-computer interfaces—whose primary application is helping people with disabilities (*e.g.*, paralysis) interact with the world—the harm caused is not merely scientific but also medical, with disproportionate impact on people with disabilities.

REPRODUCIBILITY STATEMENT

All code needed to replicate the results presented here will be released upon publication. All data needed to replicate the results presented here are publicly available. The aforementioned code downloads the data.

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
