# OpenReview forum: "Randomized trials in EEG classification experiments are mandatory as   drift is pervasive"
_ICLR.cc/2026/Conference — Submitted to ICLR 2026_

### Official Review · Reviewer_2p4q · 2025-10-27

**Soundness:** 1
**Presentation:** 1
**Contribution:** 2
**Rating:** 0
**Confidence:** 4

**Summary:**

The paper adresses a temporal correlation in a publicly available dataset, which leads to a confound in classification.  To demonstrate the confound, the authors repurposed the public dataset to argument that without this confound the classification results on this dataset have lower performance.  The authors gave an overview of articles using this dataset, that may have been affected by this confound.

**Strengths:**

- The paper gives an overview of the articles where the supposedly affected dataset has been used.
- The paper explains a method which should illustrate that without the temporal correlation, the classification results are

**Weaknesses:**

- The nature of the confound is not clearly presented.  The authors state this in the first 2 sentences in the first paragraph of section 3, but this is the key to this article. Yet, the nature of the confound is only described in 2 sentences.  This creates little convincing value in their case.  The authors could use other references to experimental designs where the confound did not appear.
- The authors put evidence of the temporal correlation by repurposing three other datasets towards the design of Spampinato et al, but I question if this is really representative.  The three datasets used in the article do not have a visual attention task, hence the question is if the LSTM network and the EEGChannelNet classifier is well suited for this data.

- The authors did not clearly explain why the filtering method would decrease the temporal correlation.

- The paper focusses on the flaw in the article but lacks a good practices statement or lessons learned discussion on how to avoid these confounds in later studies.

- The authors also tried to place an analogy with radio broadcast, but there it is not clear what the link is with EEG signals.  The paper lacks an small and consice overview on how to remove temporal correlation.

**Questions:**

- Why was the bandpass filtering between 14 and 71 Hz chosen?

- What is the effect of the post hoc filtereing?  Is the temporal correlation decreased or has it remained the same?

- Did the authors try other means to demonstrate this temporal correlation?.

- In the discussion section, the authors claim that bandpass filtering dos not remove slow spectral change, but what do the authors mean by slow.  Can they give a time estimation?  Do you mean slow spectral change outside of the bandpass limits of the filter?

---

> ### Author Response · Authors · 2025-11-22
>
> W1: Li et al. (2021), Ahmed et al. (2021, 2022), Bharadwaj et al
> (2023) discuss the confound in detail. The sole purpose of this
> manuscript is the novel demonstration that all EEG data suffers from
> drift that can lead to a confound with block designs such as those
> that are used by over 100 papers. Li et al. (2021) and Ahmed et
> al. (2021) discuss experimental designs that avoid the confound
> (randomized trials). The data from those papers are publicly available
> in IEEE Dataport. Those papers contain URLs for those datasets.
>
> W2: The whole idea of this paper is to show that one can recover
> "class" that isn't there, thus refuting Spampinato et al. (2017),
> Kavasidis et al. (2017), and Palazzo et al. (2018, 2020a, 2020b, 2021,
> 2024). Li et al. (2021) and Ahmed et al. (2021) collect data with
> visual stimuli. Li et al. (2021), Ahmed et al. (2021, 2022), Bharadwaj
> et al. (2023) all present methods that can decode that visual
> stimulus-related information. They all show that LSTM and EEGChannelNet
> cannot. The whole purpose of this paper is to refute the claims from
> Spampinato et al. (2017), Kavasidis et al. (2017), and Palazzo et
> al. (2018, 2020a, 2020b, 2021, 2024) that LSTM and EEGChannelNet
> can.
>
> W3: We never claim  that filtering decreases the confound. Palazzo et
> al. (2018, 2020b, 2021, 2024) claim that it can. Here, we refute that
> claim.
>
> W4: lines 316-318 The only known way of preventing temporal
> correlation from introducing a confound is through appropriate
> randomization that breaks the temporal correlation.
>
> W5: The idea of radio broadcast is a not an analogy. It is a
> technical argument of why filtering cannot, in principle, remove the
> confound. AM radio (actually any kind of modulation) recovers the
> signal, despite that passband of the tuner. The very last sentence of
> the paper tells how to avoid the confound
>
> Q1: Because that is what Spampinato et al. (2017), Kavasidis et
> al. (2017), and Palazzo et al. (2018, 2020a, 2020b, 2021, 2024) used
> and claim removes the confound.
>
> Q2: The whole result of this manuscript is to show that filtering does
> not remove the temporal confound.
>
> Q3: Yes. See  Li et al. (2021), Ahmed et al. (2021, 2022), Bharadwaj et al
> (2023) and submission 4802.
>
> Q4: This experiment shows that drift is manifest at the rate of stimulus
> presentation (2 Hz in the case of data from Spampinato et
> al. 2017). The sole purpose of this manuscript is to refute Spampinato
> et al. (2017), Kavasidis et al. (2017), and Palazzo et al. (2018,
> 2020a, 2020b, 2021, 2024). It shows that the 14-71Hz passband that
> they employ does not remove the drift.

---

### Official Review · Reviewer_ocZT · 2025-10-31

**Soundness:** 3
**Presentation:** 2
**Contribution:** 2
**Rating:** 4
**Confidence:** 4

**Summary:**

The manuscript highlights a critical methodological limitation pervasive in the EEG classification literature, as typified by Spampinato et al. (2017). The authors contend that inherent temporal drift in EEG recordings, when combined with block-design protocols that are inadequately randomized, produces spurious correlations between stimulus categories and the temporal structure of the experiment. This confounding factor fundamentally undermines the interpretability of results and casts doubt on the validity of conclusions drawn in nearly one hundred related publications.

**Strengths:**

- This paper demonstrated temporal correlations among three distinct public EEG datasets: Delorme (2020), Hatlestad-Hall et al. (2021), and Babayan et al. (2019). This supports the perspective that temporal correlations are widely present in EEG data.
- The results clearly demonstrate that bandpass filtering (14–71 Hz) does not remove the temporal correlation. Even with filtering, the classification accuracy obtained by both the LSTM and EEGChannelNet classifiers remains far above chance, refuting previous claims (Claim II) that filtering could remove the temporal correlation.
- The paper offers the novel and significant claim that the collection protocol used in studies like Spampinato et al. (2017) is inherently and irreparably confounded.

**Weaknesses:**

- The authors state explicitly that they do not claim the presence of confounds in the original studies corresponding to the datasets analyzed. Their approach instead repurposes three publicly available EEG datasets, comprising two resting-state and one auditory dataset, in order to reproduce the confounded block-design structure reported by Spampinato et al. (2017).
- The consistent alignment between training and test loss or accuracy observed throughout training is presented as strong evidence for the existence of label leakage.
- The high classification accuracy obtained, nearly perfect without filtering and substantially exceeding chance levels even after filtering, suggests that the classifiers examined, namely LSTM and EEGChannelNet, primarily exploit temporal confounds rather than encoding stimulus-related neural activity.

**Questions:**

- Q1. Since the data from all participants were combined before being divided into training and test sets, has any analysis been conducted, or is supporting evidence available, that demonstrates the presence of temporal correlation when subject pooling is avoided? Furthermore, is there empirical verification that the confound mechanism generalizes across individual subjects or recording sessions?
- Q2. This study provides clear evidence that applying a 14–71 Hz bandpass filter does not effectively mitigate the issue. In light of the electrical engineering principle concerning filtering and modulation cited in the paper, were alternative high-pass or band-stop filter thresholds tested to determine the precise conditions under which temporal correlation remains and cannot be eliminated after processing?
- Q3. The findings indicate that after filtering, EEGChannelNet achieves substantially higher classification accuracy than LSTM, leading to the interpretation that EEGChannelNet is even more sensitive to the confound. What analyses or theoretical considerations account for the greater susceptibility of EEGChannelNet compared to LSTM to the temporal confound under filtered conditions?

**Details Of Ethics Concerns:**

Although the accuracy and content of this refutation paper are evaluated independently, there appears to be a concern regarding potential duplicate or salami-sliced submissions.

Specifically, papers #4790 and #4793 present highly similar refutation claims, with Section 2 (Significance) showing nearly identical content. Based on this overlap, it is reasonable to suspect that the same author may have submitted multiple closely related manuscripts.

Moreover, considering reviewer 2p4q’s official comment on paper #4793 and the textual similarity observed in Section 2 of #4802, it seems likely that papers #4790, #4793, and #4802 originate from the same author.

I respectfully ask the ACs and SPCs to review this matter carefully.

---

> ### Author Response · Authors · 2025-11-22
>
> Re weaknesses:
> We are confused. We believe that all weaknesses listed by the reviewer
> are weaknesses in the work we refute (Spampinato et al. 2017,
> Kavasidis et al. 2017, Palazzo et al. 2018, 2020a, 2020b, 2021, 2024).
> They are strengths of our submission.
>
> Q1: Yes. See Li et al. (2021), Ahmed et al. (2021, 2022), Bharadwaj et
> al. (2023). Also see submission 4802.
>
> Q2: We only apply this particular passband because this is precisely
> the filtering employed in Spampinato et al. (2017), Kavasidis et
> al. (2017), and Palazzo et al. (2018, 2020a, 2020b, 2021, 2024). In
> some sense it is vacuous to ask whether one can remove temporal
> confounds can be removed by some kind of filtering. Obviously one can
> filter out the entire signal and no confound remains. The operative
> question is whether one can filter out just the confound and leave
> just the stimulus-related information in the signal. But answering
> that question requires knowing precisely what the stimulus-related
> information is in the signal, the central question in neuroscience, as
> yet unanswered. In all of science we seek to avoid confounds at the
> source by proper experiment design, not to remove them post hoc.
>
> Q3: Ahmed et al. (2021, 2022) and Bharadwaj et al. (2023), as well as
> submission 4802, fail to find any situation where either LSTM or
> EEGChannelNet achieve above-chance accuracy, either filtered or
> unfiltered, on randomized trials. Simply stated, we have no evidence
> that either can decode any stimulus-related information whatsoever. In
> light of this, it does not seem relevant to us which is more
> susceptible to confounds. The only valid solution is to collect
> confounded data in the first place.

---

### Official Review · Reviewer_gmYn · 2025-11-01

**Soundness:** 2
**Presentation:** 1
**Contribution:** 2
**Rating:** 2
**Confidence:** 3

**Summary:**

This paper analyzes three public EEG datasets repurposed with simulated block-design visual stimulus trials, showing pervasive temporal correlations that cause confounds in classification. It finds that filtering does not remove these correlations and concludes that only randomized trial designs can avoid such confounds.

**Strengths:**

The concern identified in about temporal confounds in EEG classification experiments is apt and essential to communicate to the scientific community, because such confounds fundamentally undermine the validity of a large body of published EEG decoding research, potentially misguiding future methods development and applications.

Addressing this issue is critical for ensuring methodological rigor, truthful interpretation of results, and the ethical advancement of EEG-based neuroscience and brain-computer interface research.

**Weaknesses:**

The language is generally direct but sometimes overly assertive in tone, bordering on confrontational regarding the ongoing use of flawed datasets in the community.

Detailed methodology for data repurposing and subject simulation could be clearer, possibly aided by schematics. The current method description is difficult to follow, requiring a line-by-line breakdown.

Validation would be stronger with direct experiments or replication on actual randomized visual EEG data rather than entirely simulated repurposed datasets.

**Questions:**

It would be important for the readers to understand how representative the simulated resting-state and auditory EEG data are for visual stimulus experiments. A discussion or motivation could be helpful to clarify this.

**Details Of Ethics Concerns:**

Agreeing with the concern raised earlier by reviewersm, the articles 4793 and 4802 are very similar. The two papers strongly overlap in thematic concerns and call for randomized designs but differ in specific emphases: one paper is a detailed experimental replication and refutation targeting a widely used protocol and its dataset, while the second is a broader scope showing pervasiveness of temporal correlation across datasets and disproving filtering as a solution. Both could have been combined given the context and text including the citations are highly similar.

---

> ### Author Response · Authors · 2025-11-22
>
> Re tone:
> We believe that nothing in the abstract or in sections 1, 3-6, the
> technical body of the manuscript is direct, assertive, or
> confrontational.
>
> Section 2 might appear as direct, assertive, or confrontational. But
> we believe that this is merited in response to the direct, assertive,
> and confrontational tone (not to mention numerous false, misleading,
> and unfounded statements) in Palazzo et al. (2024). The whole purpose
> of section 2 is to bring to the community's attention an egregious
> fact: despite publication of Li et al. (2021), Ahmed et al. (2021,
> 2022), and Bharadway (2023), the community has published, and still
> continues to submit and publish flawed work (over 100 papers), even
> in/to venues like NeurIPS and ICLR.
>
> Re methodology:
> At a high-level, the details don't matter. What matters is that
> Palazzo et al. (2020b, 2024) claim that their dataset does not suffer
> from a substantial temporal confound and that any confound that might
> be present can be removed by filtering. We provide evidence that all
> EEG datasets exhibit drift by demonstrating a temporal confound
> attributed to drift in three different datasets collected in three
> different labs.
>
> The way this is done is conceptually quite simple. Spampinato et
> al. (2017) collected a dataset where all and only samples (in both the
> training and test sets) of a given class were collected temporally
> adjacent in single block. We take temporally adjacent samples from EEG
> where there is no stimulus and thus no class information, and
> construct a dataset similar to that of Spampinato et al. (2017) and
> recover "class" information that isn't there. The technical details of
> how this is done is written in the style of a standard EEG paper and
> would be easily comprehensible to anyone well versed in EEG who wished
> to replicate the experiment.
>
> Re question:
> We took data from three different labs. All data is publicly available
> from openneuro.org. We sought data that lacked stimuli. We used all
> suitable data available from openneuro.org. It doesn't really matter
> that some of the data is resting state and some is from stimuli
> consisting of a single constant auditory tone. What is important is
> that the data was collected in the absence of stimulus-class
> information so classifiers that purport to recover stimulus class must
> instead be recovering something else (e.g. drift) present in the EEG
> signal.

---

### Official Review · Reviewer_zSQu · 2025-11-06

**Soundness:** 2
**Presentation:** 1
**Contribution:** 1
**Rating:** 2
**Confidence:** 2

**Summary:**

There seems to be a strong similarity between submissions 4793 and 4802. Both submissions point out a weakness of the experimental design of prior work studying visual stimuli decoding from EEG. While this reviewer agrees that randomized stimuli designs are critical in data collection to reduce the impact of confounding factors like temporal drifts often encountered in biosignals, it seems very likely that both articles originate from the same authors.

**Strengths:**

N/A

**Weaknesses:**

N/A

**Questions:**

N/A

**Details Of Ethics Concerns:**

To this reviewer, it seems likely that the authors submitted essentially the same work (submission 4793 and 4802) through different channels in hopes of achieving acceptance. While the authors might have a valid scientific critique, this method of presentation is not acceptable.

---

### Author Response · Authors · 2025-11-22
**Historical Background and Significance**

To understand this work's significance, consider this brief historical
overview.

Spampinato et al. (2017) introduced a block-designed dataset
("Perceive") and methods that claim to achieve extremely high accuracy
decoding stimulus image class from EEG recordings. This was amplified
by follow on papers (Kavasidis et al. 2017, Palazzo et al. 2018,
2020a, 2020b, 2021), many of which claim to do things like reconstruct
stimulus images from EEG recordings. Further, Tirupattur (2018) does
this with a fresh dataset (Kumar 2018) that has the same block design.

Li et al. (2021) debunked all of the above, demonstrating that the
Perceive dataset suffers from a block confound. EEG exhibits drift,
encoding a clock in the signal. Since Perceive was collected with all
and only stimuli of the same class being temporally adjacent, the
classifier can mistakenly classify the clock/drift instead of the
stimulus-related EEG response. Follow on papers (Ahmed et al. 2021,
2022, Bharadwaj et al. 2023) added novel independent confirmation of
the results of Li et al. (2021).

Despite this, Palazzo et al. (2020b, 2021, 2024) continue to argue
that their dataset is valid. At this point, there are over one hundred
papers that use the Perceive dataset, the Kumar (2018) dataset, or
other datasets that suffer from the same block confound. Many new
datasets have been collected with this same block confound, some of
which are becoming widely used. The vast majority of these were
published after the confound became known (Li et al. 2021). Some of
these are unaware of the confound. Others are aware, but dismiss it,
often based on the argument of Palazzo et al. (2020b, 2021, 2024).

That argument is what this manuscript refutes.

This confound has been extensively debated on blogs like reddit, yet
that too has not stopped the extensive publication of flawed work.

There are three distinct levels of severity of this issue, which
progressively support greater need for continued publication:

 1. Many authors are unaware of the confound, despite the fact that it
    was published in prominent venues (e.g., TPAMI, CVPR, NeurIPS) and
    continue to publish flawed work

 2. While many authors are aware of the confound, they nonetheless
    ignore the warning and continue to publish flawed work.

 3. Some authors dismiss the confound and actively argue for the
    community to continue to employ flawed methods.

---

> ### Author Response · Authors · 2025-11-22
> **Re: Concerns about duplicate submissions**
>
> We submitted five manuscripts to ICLR 2026. To summarize:
>
>   4790: Palazzo (2020b, 2021) introduces a method that claims to
>         jointly train two mappings, from EEG and images, to a common
>         embedding space. We debunk central claims about this
>         embedding. We do this both for the confounded dataset and
>         nonconfounded datasets.
>
>   4793: Palazzo et al. (2020b, 2021, 2024) claim that their dataset
>         does not suffer from drift. We show that three other datasets,
>         all collected by different people in different labs, suffer
>         from drift, demonstrating that drift is unavoidable with EEG.
>
>   4796: Palazzo et al. (2020b, 2021) produce activation maps and claim
>         that these are consistent with neuroscience knowledge. We
>         debunk this claim.
>
>   4802: Palazzo et al. (2024) further claim that their dataset is
>         valid by arguing that the experiments in Li et al. (2021),
>         Ahmed et al. (2021, 2022), and Bharadwaj (2023) were
>         improperly conducted. We repeat the experiment in Spampinato
>         et al. (2017) exactly, in a controlled fashion, where the only
>         thing varied is block order. this conclusively demonstrates
>         that Spampinato et al. (2017), Kavasidis et al. (2017),
>         Palazzo et al. (2018, 2020a, 2020b, 2021, 2024), and the one
>         hundred other papers are flawed.
>
>   4804: Palazzo et al. (2024) makes numerous false statements about
>         Li et al. (2021), Ahmed et al. (2021, 2022), and Bharadwaj (2023).
>         We correct those statements.
>
> These are all independent. There is no duplicate substantive material
> between these five submissions and Li et al. (2021), Ahmed et al.
> (2021, 2022), and Bharadwaj (2023). While they all comment on the same
> body of flawed work, they each introduce and discuss distinct
> technical issues and make distinct contributions.
>
> We included §2 Significance in all five manuscript. While largely the
> same text, this is not the technical contribution of each respective
> manuscript. It solely serves to highlight the significance of the
> specific technical contribution in each individual manuscript, namely
> that each offers independent refutation of one hundred papers. This is
> important, because even if one were to remedy one of the flaws, many
> others remain, and a large and growing corpus of work remains flawed.
> Further, it is conceivable that in the future, a paper might suffer
> from one flaw but not the other, yet it would still be invalid.

---

> > ### Author Response · Authors · 2025-11-22
> > **Re: Public debate**
> >
> > Several reviewers commented that public debate of this issue is
> > inappropriate. We realize that this may be unconventional and uncommon
> > in the ML community. But it is common in most other scientific fields
> > (e.g., Brain and Behavioral Science, Psycoloquy, ...). Public debate
> > through publication is the well-established method for arriving at
> > scientific truth. Schaeffer (2025) have argued that a mechanism for
> > publishing critiques and refutations is sorely lacking in ML.
> >
> > The vast majority of the reviews of all five of these submissions
> > focus on the fact that they are unconventional. Essentially none of
> > the reviews discuss any technical flaws in these submissions. We
> > would be happy to discuss and address any technical flaws.

---

> > > ### Author Response · Authors · 2025-11-22
> > > **Specific relevance and significance to ICLR and the ML community**
> > >
> > > It is important, if not imperative, for the community to publish this
> > > work. Without it, the community continues to submit and publish more
> > > flawed work at a growing rate. Fifty new papers papers have been
> > > published since the flaw was first reported in prominent venues: once
> > > in CVPR (Ahmed et al. 2021) and three times in TPAMI (Li at al. 2021,
> > > Ahmed et al. 2022, Bharadwaj et al 2023).
> > >
> > > Some recent flawed work has been published even by the ML community in
> > > top ML venues, despite awareness of the issue: Liu et al. (2024) in
> > > NeurIPS collects a new dataset that suffers from the block
> > > confound. While the authors cite Li et al. (2021) and Ahmed et al
> > > (2021), they fail to appreciate (or maybe hide the fact) that their
> > > work is confounded.
> > >
> > > Some recent flawed work has even been submitted to ICLR 2025 (and
> > > apparently resubmitted to ICLR 2026 despite reviewer warnings). It
> > > appears that even the reviewer pool of ICLR is unaware of the severity
> > > of the confound.
> > >
> > > https://openreview.net/forum?id=ejVuTFFkl6&noteId=zafmRtlFw1
> > >
> > > collects a new dataset that suffers from the block confound. While the
> > > authors again cite Li et al. (2021), they incorrectly claim that their
> > > dataset does not suffer from the confound. All four of the reviewers
> > > point this out. While this submission was rejected, three of the
> > > reviewers rated it as "Soundness: 3: good" and two of the reviewers
> > > rated it as "Contribution: 3: good".
> > >
> > > The apparent resubmission (18265) to ICLR 2026 again cites Li et
> > > al. (2021) and again incorrectly claims that their dataset does not
> > > suffer from the confound. Again, three of the four reviewers point out
> > > that this work suffers from the block confound. And again, two of the
> > > reviewers rate this as "Soundness: 3: good", one of the reviewers
> > > rates this as "Contribution: 3: good", and one even rates this as
> > > "Contribution: 4: excellent" and recommends acceptance.
> > >
> > > We have a simple question for the reviewers, area chairs, and program
> > > chairs: If one cannot publish refutations like this in ICLR, how else
> > > do you propose we address the fact that there is a large and growing
> > > body of flawed work being published?

---

### Meta-Review · Area_Chair_TWJp · 2026-01-01

**Summary:**

Reviewers broadly agreed that the paper targets an important and timely methodological issue about temporal drift/temporal correlation in EEG that can confound block-design decoding studies, with the reported experiments. However, they are foundemental concerns were raised about research-integrity/overlap (substantial duplication with other submissions) and insufficient clarity/validation (hard-to-follow presentation, scope/tone fit, and requests for stronger evidence of generalisation beyond the tested datasets).

**Reviewer Concerns:**

The rebuttal partially addressed concerns about motivation, tone, and the choice of filtering by providing clarifications and justifications. However, all four reviewers’ main concerns remain outstanding, particularly the duplicate-submission/research-integrity issue, as well as requests for clearer methodological exposition, stronger and more direct validation, and improved generalisation evidence, none of which were resolved with new analyses or substantive revisions.

**Reviewer Scores:**

All four reviewers gave negative scores and did not engage in the rebuttal discussion. After reading the authors’ rebuttal, I do not believe the fundamental issues raised by the reviewers have been addressed, particularly the concerns around research integrity/overlap and the lack of sufficiently clear presentation and convincing validation.

---

### Decision · Program_Chairs · 2026-01-26

Reject